



# Highly time-resolved measurements of element concentrations in PM10 and PM2.5: Comparison of Delhi, Beijing, London, and Krakow

Pragati Rai[1], Jay G. Slowik[1], Markus Furger[1], Imad El Haddad[1], Suzanne Visser[2], Yandong Tong[1], Atinderpal Singh[3], Günther Wehrle[1], Varun Kumar[1], Anna K. Tobler[1], Deepika Bhattu[1,a], Liwei Wang[1], Dilip Ganguly[4], Neeraj Rastogi[3], Ru-Jin Huang[5], Jaroslaw Necki[6], Junji Cao[5], Sachchida N. Tripathi[7], Urs Baltensperger[1], André S. H. Prévôt[1]

[1]Laboratory of Atmospheric Chemistry, Paul Scherrer Institute, Forschungsstrasse 111, 5232 Villigen PSI, Switzerland
[2]Centre for Environmental Quality, National Institute for Public Health and the Environment, 3720 BA, Bilthoven, the Netherlands
[3]Geosciences Division, Physical Research Laboratory, Ahmedabad 380009, India
[4]Centre for Atmospheric Sciences, Indian Institute of Technology Delhi, New Delhi 110016, India
[5]Key Laboratory of Aerosol Chemistry and Physics, Institute of Earth Environment, Chinese Academy of Sciences, Xi'an 710075, China
[6]Faculty of Physics and Applied Computer Science, Department of Applied Nuclear Physics, AGH University of Science and Technology, 30059 Krakow, Poland
[7]Department of Civil Engineering and Department of Earth Sciences, Indian Institute of Technology Kanpur, Kanpur, Uttar Pradesh 208016, India
[a]now at: Department of Civil and Infrastructure Engineering, Indian Institute of Technology Jodhpur, Jodhpur, Rajasthan 342037, India

*Correspondence to*: Markus Furger (markus.furger@psi.ch), André S. H. Prévôt (andre.prevot@psi.ch), S. N. Tripathi (snt@iitk.ac.in), Junji Cao (cao@loess.llqg.ac.cn), Jaroslaw Necki (necki@agh.edu.pl)

**Abstract.** We present highly time-resolved (30 to 120 min) measurements of size-fractionated (PM10 and PM2.5) elements in two cities in Asia (Delhi and Beijing) and Europe (Krakow and London). For most elements, the mean concentrations in PM10 and PM2.5 are higher in Asian cities (up to 24 and 28 times, respectively) than in Krakow, and often higher in Delhi than in Beijing. Among European cities, Krakow shows higher elemental concentrations (up to 20 and 27 times, respectively) than London. The enrichment factor of an element together with the size distribution allows for a rough classification of elements by major sources. We define five groups: (1) dust-related, (2) non-exhaust traffic emissions, (3) solid fuel combustion, (4) mixed traffic/industrial emissions, and (5) industrial/coal/waste burning emissions, with the last group exhibiting the most site-to-site variability. Hourly maximum concentrations of Pb and Zn reach up to 1 µg m⁻³ in Delhi, substantially higher than at the other sites. We demonstrate that the high time resolution and size-segregated elemental dataset can be a powerful tool to assess aerosol composition and sources in urban environments. Our results highlight the need to consider the size distributions of toxic elements, diurnal patterns of targeted emissions, and local vs. regional effects in formulating effective environmental policies to protect public health.

## 1 Introduction

The percentage of the global population living in urban areas with more than 1 million inhabitants has been steadily increasing over the last decades (Krzyzanowski et al., 2014). Air pollution in these cities is a major contributor to the global disease burden (Lim et al., 2012), with more than 96% of the population in these cities exposed to PM2.5 (particulate matter with an aerodynamic diameter below 2.5 µm) above World Health Organization (WHO) air quality standards (Krzyzanowski et al., 2014). Smaller particles are likely more toxic since they can penetrate deep into the lungs (Miller et al., 1979). Particle toxicity depends also on PM composition (Kelly and Fussell, 2012), with identified toxic constituents including elemental and organic carbon, and metals. Transition metals such as Fe, V, Ni, Cr$^{VI}$, Cu and Zn are of particular concern due to their potential to produce reactive oxygen species (ROS) in biological tissue (Manke et al., 2013). Moreover, metals such as Pb, Cd and the metalloid As accumulate in body tissue and contribute to many adverse health effects, such as lung cancer, cognitive deficits,





and hearing impairment (Jaishankar et al., 2014). Elements are also recognized as effective markers for source apportionment
(SA), especially for anthropogenic emissions in urban areas (e.g., traffic, industry and power production). Emissions from
these sources vary on timescales of a few hours or less, and such rapid changes cannot be resolved by conventional 24-h filter
measurements. The vast majority of elemental SA studies in the literature are limited by the time resolution of the input samples
(Dall'Osto et al., 2013; Pant and Harrison, 2012). Highly time-resolved and size-segregated measurements are thus required
for the determination of elemental PM sources and health effects within urban areas under varying meteorological conditions.

Efforts in European and Asian countries to tackle poor air quality include the EURO norms (EEA, 2018) in European cities to
control vehicular emissions, odd-even traffic regulations in Delhi (Kumar et al., 2017) and Beijing (An et al., 2019), and the
"stop smog" program in Poland (Shah, 2018). In addition, strict emission control measures were implemented in China (Gao
et al., 2016) in September 2013, by lowering the fraction of coal in energy production from 24 % in 2012 to 10 % in 2017.
Evaluation and optimization of such programs require elucidation of the sources and processes governing PM abundance and

composition. This remains challenging and may strongly differ from site to site depending on local environmental conditions.
To assess this, we present high time resolution $PM_{10}$ and $PM_{2.5}$ metal and trace element concentrations in four Asian and
European cities: Delhi, Beijing, Krakow, and London. A simple conceptual framework allows characterization of major
sources, site-to-site similarities and local differences, and identification of key information required for efficient policy
development. Moreover, this method does not requiring a full source apportionment (SA) analysis (presented elsewhere for

London and Delhi (Visser et al., 2015a; Rai et al., 2020)), which is complex and time-consuming, and which can be challenging
to compare across sites due to differences in source definitions.

## 2 Materials and Methods

### 2.1 Description of the campaigns

The sampling site (40.00° N, 116.38° E) in Beijing was located in a residential area north of the urban core, near the Olympic
Park without any nearby industrial sources. It is a typical urban site in the central zone of Beijing. It is located approximately
1.2 km away from the west 3$^{rd}$ Ring Road and 2.7 km away from the north 2$^{nd}$ Ring Road. Both ring roads are characterized
by heavy traffic. Coal-based heating is a major sector of coal consumption in Northern China (Tian et al., 2015). The
measurements were performed from 6 November to 12 December 2017.

The sampling site (50.06° N, 19.91° E) in Krakow was located in a residential area close to the city center. The major local
sources of pollution are municipal emissions, combustion, industry, and traffic. Traffic in the city is dense with frequent traffic
jams (~1 km away from sampling location). Factories (steel and non-ferrous metallurgical industries) are located at a distance
of about 10 km from the sampling site. Additionally, a coal power plant is located in the southern area of the city. Moreover,
the zinc ore industry is situated about 50 km to the north of the city. The sources with the highest PM emission rates are situated
in the northeastern part of Krakow, i.e., Huta Arcelor Mittal steel works, the Cementownia cement factory and the EC Krakow
coal-fired power plant (Junninen et al., 2009). However, in Krakow, there are numerous small coal-fired low-efficiency boilers
(LE-boilers) distributed over the city. The measurements were performed from 11 to 23 October 2018.

The Delhi sampling location (28.54° N, 77.19° E) was situated in a residential and commercial area in the south part of Delhi.
Roads with heavy traffic within 2-5 km surround the sampling location in all directions. Many anthropogenic sources such as
traffic, agricultural residue burning, waste burning, one coal based power plant, various micro-, small-, and medium-scale
manufacturing and processing units such as metal processing, electroplating, and paint and chemical manufacturing for pre-
treatment of metals, might contribute to the low air quality of this region. However, the coal based power plant in the southeast
direction (18 km) was shut down in October 2018, although evacuation of fly ash continued during the study period. The
measurements were performed from 15 January to 9 February 2019.



The London sampling location (51.52° N, 0.21° W) classified as urban background, was within a school ground in a residential
area of North Kensington (NK). Long-term measurements of air pollutants at NK have been described in detail in a previous
study (Bigi and Harrison, 2010), and are considered as representative of the background air quality for most of London. NK is
situated within a heavy traffic suburban area of London. The measurements were performed from 6 January to 11 February
2012.

### 2.2 Instrumentation

In Beijing, Delhi, and Krakow, sampling and analysis was conducted with an Xact 625i® Ambient Metals Monitor (Cooper
Environmental, Tigard, OR, USA) with an alternating $PM_{10}$ and $PM_{2.5}$ inlet switching system (Furger et al., 2020). Details of
the Xact can be found in previous studies (Cooper et al., 2010; Furger et al., 2017; Rai et al., 2020; Tremper et al., 2018). The
field measurements with the Xact were performed with 1 h time resolution in Beijing and 0.5 h time resolution in Krakow and
Delhi. The instrument was able to detect 34 elements (Al, Si, P, S, Cl, K, Ca, Ti, V, Cr, Mn, Fe, Co, Ni, Cu, Zn, Ga, Ge, As,
Se, Br, Rb, Sr, Y, Zr, Cd, In, Sn, Sb, Ba, Hg, Tl, Pb and Bi). However, some of the elements were below minimum detection
limit (MDL) of the instrument (Table S1) for certain periods of time. Therefore, we discarded the elements that were below
MDL in $PM_{10}$ and $PM_{2.5} \geq 80\%$ of the time.

In London, we deployed a rotating drum impactor (RDI) which sampled with 2 h time resolution in size-segregated stages:
$PM_{10-2.5}$ (coarse), $PM_{2.5-1.0}$ (intermediate) and $PM_{1.0-0.3}$ (fine). Trace element composition of the RDI samples was determined
by synchrotron radiation-induced X-ray fluorescence spectrometry (SR-XRF) at the X05DA beamline (Flechsig et al., 2009)
at the Swiss Light Source (SLS), Paul Scherrer Institute (PSI), Villigen PSI, Switzerland, and at Beamline L at the Hamburger
Synchrotronstrahlungslabor (HASYLAB), Deutsches Elektronen-Synchrotron (DESY), Hamburg, Germany (beamline
dismantled in November 2012). In total 25 elements were quantified (Na, Mg, Al, Si, P, S, Cl, K, Ca, Ti, V, Cr, Mn, Fe, Ni,
Cu, Zn, Br, Sr, Zr, Mo, Sn, Sb, Ba, Pb). Details of the RDI-SR-XRF analysis were described in previous studies (Bukowiecki
et al., 2008; Richard et al., 2010; Visser et al., 2015b). Due to the RDI's omission of particles smaller than 300 nm, the fine
mode elemental data for London is less reliable as compared to other sites. While the comparison of size-resolved London data
with the other sites should therefore be interpreted with caution, we present London $PM_{2.5}/PM_{10}$ ratios, group classification (in
$PM_{10}$ and $PM_{2.5}$) and their diurnal patterns (in $PM_{2.5}$ and coarse ($PM_{10}$-$PM_{2.5}$)) in the Supplement (Figs. S4, S5 and S8,
respectively).
Xact measurements of Cl and S were compared to the chloride and sulfate data obtained from co-located aerosol mass
spectrometer (AMS) measurements (Fig. S1). The AMS instruments consisted of a high-resolution long-time-of-flight (L-
TOF) AMS deployed for online measurements of size segregated mass spectra of non-refractory (NR)-$PM_{2.5}$ with 2 min
resolution in Beijing and a HR-TOF-AMS of NR-$PM_1$ with 2 min resolution in Delhi. The scatter plots exhibit a good
correlation, which is reflected by a Pearson's $R$ of 0.91 (Delhi) and 0.96 (Beijing) for S vs sulfate, and 0.98 (Delhi) and 0.97
(Beijing) for Cl vs chloride. The correlation resulted in a slope of 1.13 (Delhi) and 1.23 (Beijing) for sulfate, and 1.03 (Delhi)
and 1.9 (Beijing) for Cl. The S measurements of the two instruments agree within the typical uncertainties of such
measurements (~25%) (Canagaratna et al., 2007; Furger et al., 2017). In addition, the Delhi measurements cover different size
fractions ($PM_{2.5}$ for the Xact vs. $PM_1$ for the AMS). The Xact/AMS ratio for Cl observed in Beijing likely occurs because the
relative ionization efficiency for AMS measurements of Cl was not determined in Beijing (whereas calibrations with $NH_4Cl$
were performed in Delhi). In addition, the Beijing measurements likely have a higher fraction of other forms of Cl (e.g. $ZnCl_2$,
$PbCl_2$, $FeCl_3$), which are not efficiently detected in standard AMS operation. Despite these uncertainties in the absolute AMS
Cl concentrations in Beijing, the two methods are highly correlated, suggesting good data quality.





### 2.3 Crustal enrichment factor (EF) analysis

EF analysis was applied to determine the enrichment of a given element relative to its abundance in the upper continental crust
(UCC). For this analysis Ti (Fomba et al., 2013; Majewski and Rogula-Kozłowska, 2016; Wei et al., 1999) was selected as the
reference element due to its stable and spatially homogenous characteristics in the soil. The compilation of UCC (Rudnik and
Gao, 2003) was used to calculate EFs and crustal contributions on elemental concentrations. For an element (X) in a sample,
the EF relative to Ti is given as:

$$EF = \frac{(X/Ti)_{Sample}}{(X/Ti)_{Crust}} \tag{1}$$

The unexpectedly low EFs observed for Si (0.41–0.45) and compared to previous studies (Majewski and Rogula-Kozłowska,
2016; Tao et al., 2013), are likely due to differences in the soil composition relative to the assumed values for the continental
crust. Given that Si is the only outlier across all measured elements, a major anthropogenic contribution to Ti seems unlikely.
However, Ti emission is possible from non-exhaust traffic sources, measured in road dust samples worldwide (Amato et al.,
2009; Pant et al., 2015).

## 3 Results

### 3.1 PM$_{10el}$ and PM$_{2.5el}$ concentration

Hourly average elemental PM$_{10}$ (PM$_{10el}$) and elemental PM$_{2.5}$ (PM$_{2.5el}$) concentrations were measured, where Figure 1 (a, b)
summarizes the results of 18 elements measured at all four sites. Total measured concentrations at Delhi (54 µg m$^{-3}$ in PM$_{10}$;
32 µg m$^{-3}$ in PM$_{2.5}$) are three times higher than those at the other sites, followed by Beijing (16.7 µg m$^{-3}$; 5.2 µg m$^{-3}$), Krakow
(9 µg m$^{-3}$; 4.3 µg m$^{-3}$) and London (1.9 µg m$^{-3}$; 0.9 µg m$^{-3}$) (Fig. S2a). Although the measurement periods do not overlap, they
were all performed during the colder months of the year (see Section 2.1), and characteristic features of each site are evident.
The total PM$_{10el}$ and PM$_{2.5el}$ concentrations in Delhi show a strong diurnal cycle, with high concentrations overnight and in the
early morning hours, followed by a sharp decrease during the day (Fig. S2b for time series, Fig. 4 for PM$_{10el}$ diurnal cycle). In
contrast, Beijing experiences multi-day haze events with only minor diurnal cycling (Fig. S3). In Krakow and London,
concentrations are mostly elevated during the rush-hours and during daytime in general (from 08:00 until 18:00 local time
(LT)).

At all four sites, Si, Cl, Fe, S, Ca, and K account for >95% of PM$_{10}$ (>88% without K) and >94% of PM$_{2.5}$ (see Fig. 1b, Tables
S2 and S3). Among elements with higher atomic numbers (Z= 29–82), Zn and Pb are highest at all sites except London (where
Zn and Cu show the highest concentrations). Figure 1d presents the mean PM$_{10el}$ concentrations normalized to those in Krakow.
With rare exceptions, element concentrations were highest in Delhi followed by Beijing, Krakow, and London. The
concentrations of toxic PM$_{10el}$ (Cr, Ni, Fe, Cu, Zn, As and Pb) in Delhi are higher than at any other site, such as Cr (2 to 9
times), Ni (2 to 8 times), Mn (1 to 16 times), Cu (4 to 13 times), Zn (5 to 95 times) and Pb (12 to 205 times). However, the
mean concentrations of carcinogenic elements (Pb, Ni, As, and Cr) (IARC, 2020) fall below the US EPA recommended
inhalation reference concentrations (RfC) for resident air (200 ng m$^{-3}$, 20 ng m$^{-3}$, 15 ng m$^{-3}$, and 100 ng m$^{-3}$, respectively)
(USEPA, 2020) except for Pb in Delhi, which exceeds the RfC by more than a factor of 2. Individual exceedances of the RfC
are relatively common in Delhi for Pb (52.8% of data) and As (34%), indicating severe risks to human health. At other sites,
RfC exceedances are less common, comprising only 10% of As data in Beijing, and 1.76% of Cr and 1.4% of Ni in Krakow;
no other RfC exceedances are observed.

### 3.2 Characteristic element groups

To evaluate the similarities and differences in element behaviour across sites, we investigate the PM$_{10}$ enrichment factor (EF)
for each element and their corresponding PM$_{2.5}$ to PM$_{10}$ ratios where EFs >> 1 indicate strong anthropogenic influence. In



addition, the particle size distribution, represented here as the mass ratio $PM_{2.5}/PM_{10}$ for an element, reflects the corresponding emission processes and can provide insight into specific sources. For example, abrasion (e.g., mineral dust and brake wear) results in coarse particles, whereas combustion and industrial processes are more likely to emit fine particles.

Figure 2 shows the $PM_{10}$ EFs as a function of $PM_{2.5}/PM_{10}$ for all elements measured at Delhi, Beijing, and Krakow (see Fig. S5 for London). Each site is shown separately in Fig. 2 and overlaid in Fig. S5. $PM_{10}$ EFs for all sites and $PM_{2.5}/PM_{10}$ for Delhi, Beijing, and Krakow are shown in Fig. 1c and Fig. 3 (see Fig. S4 for London together with other sites), respectively. In general, EFs increase with increasing $PM_{2.5}/PM_{10}$. From Fig. 2, we divide the measured elements into 5 groups based on their position in the EF vs. $PM_{2.5}/PM_{10}$ space; this framework provides insight into element sources and emission characteristics.

The classification for London is uncertain due to the lower cut-off issue mentioned in Section 2.2, but some qualitative agreement with the other sites is evident, with the largest differences related to the $PM_{2.5}/PM_{10}$ ratio. Therefore, London is included in the group classification below, although the data are shown in the Supplement for ease of viewing. Figure 4 compares the $PM_{10}$ diurnal cycles of representative elements from the five groups for all four sites normalized to the mean element concentration, while Fig. 5 compares the absolute concentrations $PM_{2.5}$ and coarse diurnals for the same elements on

a site-by-site basis for Delhi, Beijing and Krakow (See Fig. S8 for London). Diurnals of other elements are shown in Figs. S6 and S7. The groups are discussed below.

**Group 1** consists of elements with the lowest EFs and the highest fraction of coarse particles. It includes Ca, Si, and Ti at all three sites, Sr at Delhi and Beijing, and Fe in Delhi, and Zr in Beijing. Elements consistently associated with this group are typically of crustal origin, consistent with their position in Fig. 2. In contrast, Zr and Fe have been linked to both brake wear

and mineral dust in urban environments (Moreno et al., 2013; Visser et al., 2015b).

Si is selected as the Group 1 element. A strong traffic influence (i.e., rush-hour peaks) on $PM_{10}$ is evident at London, Krakow, and Delhi, while a much flatter diurnal with only small rush-hour effects is evident in Beijing (Fig. 4). $PM_{2.5}$ concentrations are very low and in general not significant relative to $PM_{10}$ (Fig. 5). These diurnal patterns are consistent with vehicle-induced resuspension of the dust deposited on the road surface, which in turn derive mostly from road abrasion, vehicle abrasion and

airborne dust from construction activities or agricultural soil.

**Group 2** elements have low EF but mean $PM_{2.5}/PM_{10}$ between 0.22 and 0.43. The increased $PM_{2.5}/PM_{10}$ value also corresponds to increased temporal variation in $PM_{2.5}/PM_{10}$, as shown by the larger interquartile range in Fig. 3. Group 2 includes Ba, Ni, Mn at all three sites, while Rb, Cr, Fe, and Zr at two sites, and Sr at a single site (Fig. 2). Several of these elements are associated with multiple sources, including coarse traffic emissions such as brake wear (e.g., Ni, Mn, Fe, Ba and Zr)

(Bukowiecki et al., 2010; Srimuruganandam and Nagendra, 2012; Visser, et al., 2015a) and other anthropogenic sources such as industrial emissions or oil burning (Ni), or crustal material (Fe and Zr).

Because of these multiple sources, several Group 2 elements show significant site-to-site variation, despite remaining in or near the group boundaries. For example, Fig. 3 shows that Ni has a similar lower quartile for $PM_{2.5}/PM_{10}$ across all sites, while the upper quartile is much higher at Krakow. This is likely due to the strong influence of local steel/non-ferrous metallurgical

industries (Samek et al., 2017a; Samek et al., 2017b), whereas the other sites are more strongly influenced by non-exhaust emissions and dust (Grigoratos and Martini, 2015; Pant and Harrison, 2012; Yu, 2013). Such differences are also evident in the Ni diurnals and time series (Figs. S6, S7 and S9), as Ni concentrations in Krakow are driven by strong isolated plumes.

As an example of a typical Group 2 element, the diurnal patterns of Ba are shown in Figs. 4, 5 and S8. Similar to Group 1, significant rush-hour peaks are evident, although the trend is now also reflected in $PM_{2.5}$. In the Asian cities, high

concentrations are also observed overnight. This is likely related to heavy-duty vehicular activities, which in these cities occur predominantly at night due to their ban during peak traffic hours (07:30 – 11:00 LT and 17:00 – 22:00 LT and less dominant during daytime) in Delhi (Rai et al., 2020) and the entire day in Beijing (Zheng et al., 2015). As the two non-exhaust traffic emissions (i.e., brake wear and dust resuspension) are related to traffic activity, the time series of most elements in Groups 1 and 2 are relatively well correlated, although not as tightly as the Group 1 elements are among themselves due to their common





source. This is illustrated in the correlation matrices shown in Fig. S10, where elements are sorted by group along each axis. Group 2 elements are also relatively well correlated among themselves at all sites, with the exception of Ni at Krakow for the reasons discussed above.

**Group 3** includes K at all three sites and adds Rb at Krakow (Fig. 2). These elements show low EF and high $PM_{2.5}/PM_{10}$, although uncertainties are high for Rb at Krakow given that 86% and 65% data points in $PM_{2.5}$ and $PM_{10}$, respectively, are

below MDL. Although coarse mode K can result from sea/road salt (Gupta et al., 2012; Zhao et al., 2015) and mineral/road dust (Rahman et al., 2011; Rogula-Kozłowska, 2016; Viana et al., 2008), the high fraction of K observed in the fine mode suggests solid fuel (coal and wood) burning as a larger source (Cheng et al., 2015; Pant and Harrison, 2012; Rogula-Kozłowska, et al., 2012; Rogula-Kozłowska, 2016; Viana et al., 2013; Waked et al., 2014). Further, Delhi, Beijing and Krakow are far from the ocean and de-icing salt was not used on the roads during the measurement periods. In London and Delhi, K was

attributed to solid fuel combustion via SA studies (Rai et al., 2020; Visser et al., 2015a). The diurnals in Delhi and Krakow show elevated values in the evening (Fig. 4), which is likewise consistent with solid fuel combustion for domestic heating. However, in Beijing only $PM_{2.5}$ exhibits such a diurnal variation (Fig. 5), whereas the $PM_{10}$ fraction is similar to the other sites without a clear diurnal variation (Fig. 4). This corresponds to a wider spread of $PM_{2.5}/PM_{10}$ at Beijing (with the lower quartile approaching values typical of Group 1), suggesting a larger contribution from dust.

**Group 4** has somewhat higher EFs than Groups 1-3 and moderate $PM_{2.5}/PM_{10}$. The group contains Cu at Beijing and Krakow, as well as Sn at Beijing and Cr at Krakow. No elements are assigned to this group in Delhi, although Cu is near the border. The EFs of these elements are >> 100 in $PM_{2.5}$ and > 10 in $PM_{10}$ (Fig. S5), indicating strong anthropogenic influence. The Group 4 elements are typically emitted from both traffic (characteristic of Group 2) and industrial or waste combustion sources (Group 5), and their position in Fig. 2 reflects the combination of these different sources. For example, Cu derives from brake

wear in Europe (Thorpe and Harrison, 2008; Visser et al., 2015a) and Asia (Iijima et al., 2007), while Cu and Sn are also emitted from industry or waste burning (Chang et al., 2018; Das et al., 2015; Fomba et al., 2014; Kumar et al., 2015; Venter et al., 2017). Cr has also been found in the emissions from both traffic (Hjortenkrans et al., 2007; Thorpe and Harrison, 2008) and oil burning in Krakow (Samek et al., 2017a).

The diurnal patterns of Cu are shown in Figs. 4, 5 and S8. London, Beijing, and Krakow all show peaks during the morning

and evening rush-hours, mainly due to the $PM_{10}$ fraction. In Krakow, $PM_{2.5}$ is approximately correlated with the coarse fraction, although the morning peak appears ~2 h later, while in Beijing $PM_{2.5}$ Cu is instead elevated at night. Delhi contrasts sharply with the other sites, which probably is the reason why Cu in Delhi is not categorized in Group 4. Figure 3 shows that the $PM_{2.5}/PM_{10}$ median and quartiles are similar, but the mean (0.72 in Delhi, and 0.46 in Beijing and Krakow) is substantially higher in Delhi because the Cu time series (Fig. S11) is subject to a series of high intensity $PM_{2.5}$ plumes from local industries

and/or waste burning. These plumes are tightly correlated with those of Cd, suggesting emissions from Cd-copper alloy manufacturing plants (Vincent and Passant, 2006), electronic waste burning (Rai et al., 2020; Owoade et al., 2015) and/or steel metallurgy (Tauler et al., 2009).

**Group 5** elements have both the highest EF and highest $PM_{2.5}/PM_{10}$ values. Similar to Groups 1-4, Group 5 includes elements that are directly emitted in the particle phase (elements mainly present in primary components), but differs by also including

elements for which the major fraction is likely emitted as gases and converted via atmospheric processing to lower volatility products which partition to the particle (elements mainly present in secondary components). Primary components elements include As, Zn, Se, and Pb at all three sites, Sn at Delhi and Krakow, and Cu in Delhi, while secondary components elements comprise Cl, Br, and S at all three sites. Although Cl and Br can in principle relate to primary emission of sea or road salt, this is unlikely for the sites studied (except London) due to the large distance from the sea, strong and regular diurnal patterns

inversely related to temperature, and correlation with elements characteristic of coal combustion and industrial emissions. In London, a major fraction of Cl was attributed to sea/road salt (Visser et al., 2015a). Further, Xact S and Cl measurements show a strong correlation with AMS-derived non-refractory $SO_4^{2-}$ and $Cl^-$, respectively, which is nearly insensitive to Cl from





sea/road-salt Cl (Fig. S1). Because the kinetics of secondary aerosol condensation are driven by surface area rather than volume, the $PM_{2.5}/PM_{10}$ of these elements is among the highest recorded, with the partial exception of Cl. In Delhi, Cl

$PM_{2.5}/PM_{10}$ values are high, consistent with a high fraction of $NH_4Cl$. However, the interquartile range of Cl $PM_{2.5}/PM_{10}$ at Beijing and Krakow is quite wide (0.5 to 0.9), with the lower values approximately matching those of Zn and Pb and suggesting that primary emissions of $ZnCl_2$ and $PbCl_2$ are not negligible at these sites.

The primary component elements of Group 5 are strongly linked to various industries and combustion of non-wood fuels. Pb was found to be present in very high concentrations in Delhi with episodic peaks, and possible sources include industrial

emissions (Sahu et al., 2011), waste incineration (Kumar et al., 2018), and small-scale Pb-battery recycling units (Jaiprakash et al., 2017). Additionally, burning of plastic and electronic waste can contribute to Pb in Delhi. Zn and As are emitted from a variety of sources, including industries, refuse burning/incineration, and coal combustion, but Zn is also emitted from traffic and wood burning. In Beijing and Krakow, coal burning from coal power plants (Samek, 2012; Yu, 2013) and domestic heating, iron and steel industries (Samek et al., 2018; Yang et al., 2013) are major sources for Zn, Se, As, and Pb. Cu and Sn

also have industrial sources, as discussed in connection with Group 4.

The set of potential sources discussed above for the primary Group 5 elements is complex and highly site-dependent, which corresponds to the significant differences between sites evident in the Group 5 correlation matrices (Fig. S10). In Beijing, Pb, Zn, Cl, Br, Se, and S are all tightly correlated, consistent with coal burning emissions. Similar correlations are observed in Krakow, with the exception of Zn and Pb, which are rather correlated with each other, as well as Mn and Fe. The Zn and Pb

time series in Krakow contain high intensity plumes (Fig. S12) with a strong peak at ~11:00 LT in $PM_{2.5}$ (Figs. 4 and 5), suggesting industrial emissions (Logiewa et al., 2020). The correlation pattern in Delhi is more complex than at the other sites, with several pairs of tightly correlated elements (e.g., Br and Cl; Se and S) but few larger groupings. This suggests plumes from a variety of point sources rather than a regionally homogeneous composition.

The location-specific influences on primary component elements in Group 5 are also evident in the diurnal patterns. For

example, as shown in Fig. 4, the diurnal pattern of Pb is relatively flat in Beijing with a slight rise in the evening, peaks approximately 08:00-10:00 LT in London, peaks at ~11:00 LT with a tail extending into the afternoon in Krakow, and has a strong diurnal cycle with a massive pre-dawn peak in Delhi. Site-to-site differences are also evident in the location of the elements within the Group 5 box in Fig. 2 (and Fig. S5). Systematic shifts are evident between Beijing (elements clustered to the lower left), Delhi (elements clustered to the upper right; note that two of the elements at the lower left are Cu and Zn,

which require a significant shift towards the upper right to even be included in Group 5), and Krakow (intermediate). This site-dependent shift contrasts with Groups 1-3, where no systematic changes are evident. Interestingly, this appears to be a feature of industrial emissions rather than anthropogenic emissions more generally, as it is not evident in the traffic or biomass combustion-dominated groups (Groups 2 and 3).

**4 Discussion and conclusions**

The broad intercontinental comparison presented here demonstrates both the large degree of similarity and crucial local differences in the $PM_{el}$ concentration and composition in European and Asian cities. The combination of $PM_{10el}$ EF and $PM_{2.5}/PM_{10}$ provides a robust and useful framework for categorizing elements and assessing site-to-site differences. Five groups are identified based on these metrics (see Fig. 2), with Groups 1-3 having low EF with increasing $PM_{2.5}/PM_{10}$ and Groups 4-5 having high EF with increasing $PM_{2.5}/PM_{10}$. Broadly, Group 1 is related to crustal materials and road dust, Group

2 to non-exhaust traffic emissions, Group 3 to biomass combustion, Group 4 to mixed industrial/traffic emissions, and Group 5 to industrial emissions and coal/waste burning. On an element-by-element basis, the group composition remains relatively consistent across sites, although some reassignment of elements occurs depending on local sources and conditions.





Interestingly, we observe systematic shifts within the EF vs. $PM_{2.5}/PM_{10}$ space only for Group 5 (and perhaps in the sparsely populated Group 4), but not in Groups 2 or 3 despite of these groups also being dominated by anthropogenic sources.

However, the consistent classification of elements into a particular group regardless of site does not imply that the temporal behavior of these elements is independent of local conditions or policies. For example, the stagnant meteorological conditions frequently encountered in Beijing during the colder season suppress diurnal variation regardless of element source, while the multitude of strongly emitting point sources yielding individual plumes in Delhi, coupled with rapid dilution as the boundary layer rises, leads to systematic, intense pre-sunrise peaks in concentration but with a composition that strongly varies on a day-

to-day basis. The effects of air quality policy are also evident, as the night to day concentration ratios of resuspension-related elements (crustal material, road dust, and non-exhaust traffic emissions) are significantly higher in Delhi and Beijing than in Krakow and London, due to time restrictions on heavy-duty truck activity in the Asian cities.

  The diurnal patterns of the total $PM_{10el}$ concentrations (Fig. 4) reflect many of the trends discussed above. Meteorological conditions yield a relatively flat diurnal pattern for Beijing, while concentrations are highest overnight and in the early morning

(before rush-hour) in Delhi due to the combined effects of industrial emissions, burning of various solid fuels, and a shallow boundary layer. Krakow and London instead have their highest $PM_{10el}$ concentrations during the day, but features related to the rush-hour are more visible in Krakow, whereas the London diurnals are similar to that of resuspended dust (Visser et al., 2015a). This may reflect differences in the fleet composition, specifically a higher fraction of older vehicles, vehicles with faulty catalytic converters or diesel particulate filters in Krakow (Majewski et al., 2018).

The global similarities and local differences discussed above should be considered in air quality policy formulation. Current practices focus mainly on total PM mass reduction, neglecting its toxicity. As an example, the carcinogenic elements represent a specific health concern. These elements are not assigned to a single group by the EF vs. $PM_{2.5}/PM_{10}$, and the group(s) to which they are assigned do not necessarily correlate with total $PM_{10el}$. While such policies may have significant ancillary benefits, they may not efficiently address the most critical health risks. In addition, the inhalability of potential toxins needs

consideration; Pb and As (which are more industry-related) have $PM_{2.5}/PM_{10}$ values that are up to 3 times higher than those of Ni and Cr (which are more traffic-related). If size dependence is not considered, inefficient or ineffective regulatory priorities may result. Finally, this study demonstrates that regulatory policy can affect not only overall concentrations but also the timing of daily maxima (e.g. truck activity restrictions in Delhi and Beijing). The above considerations highlight the importance of time- and size-resolved measurements for policy formulation, as well as the need to integrate these with daily human activities.

### Data availability


  The data presented in the text and figures as well as in the supplement will be available upon publication of the final manuscript (https://zenodo.org). Additional related data can be made available by the corresponding authors (MF and ASHP) upon request.

### Author contributions

  PR and JGS wrote the paper with input from all co-authors. PR, MF, DB, YT, VK, AKT, LW, SV, AS, JN designed the study.

GW designed ISS in Xact. YT and AP analyzed AMS data. PR analyzed Xact data. SV, MF, and JGS provided offline data for London. ASHP, JGS, MF, IEH, and UB were involved with the supervision. ASHP, JGS, MF and UB assisted in the interpretation of the results.

### Competing interests

  The authors declare no competing financial interests.





**Acknowledgements**

This study was funded by the Swiss National Science Foundation (SNSF grants 200021_162448, 200021_169787 and BSSGI0_155846), and by the Swiss Federal Office for the Environment (FOEN). We also acknowledge the Sino-Swiss Science and Technology Cooperation (SSSTC) project HAZECHINA (Haze pollution in China: Sources and atmospheric evolution of particulate matter) with SNF number IZLCZ2_169986 and NSFC number 21661132005. S.N.T. was financially
supported by the Department of Biotechnology (DBT), Government of India under grant no. BT/IN/UK/APHH/41/KB/2016-17 and by Central Pollution Control Board (CPCB), Government of India under grant no. AQM/Source apportionment EPC Project/2017. We are grateful to Jamie Berg, Krag Petterson and Varun Yadav of Cooper Environmental Services for instrument troubleshooting during field campaigns. We thank René Richter of PSI for his tremendous support for building the Xact housing and inlet switching system.

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





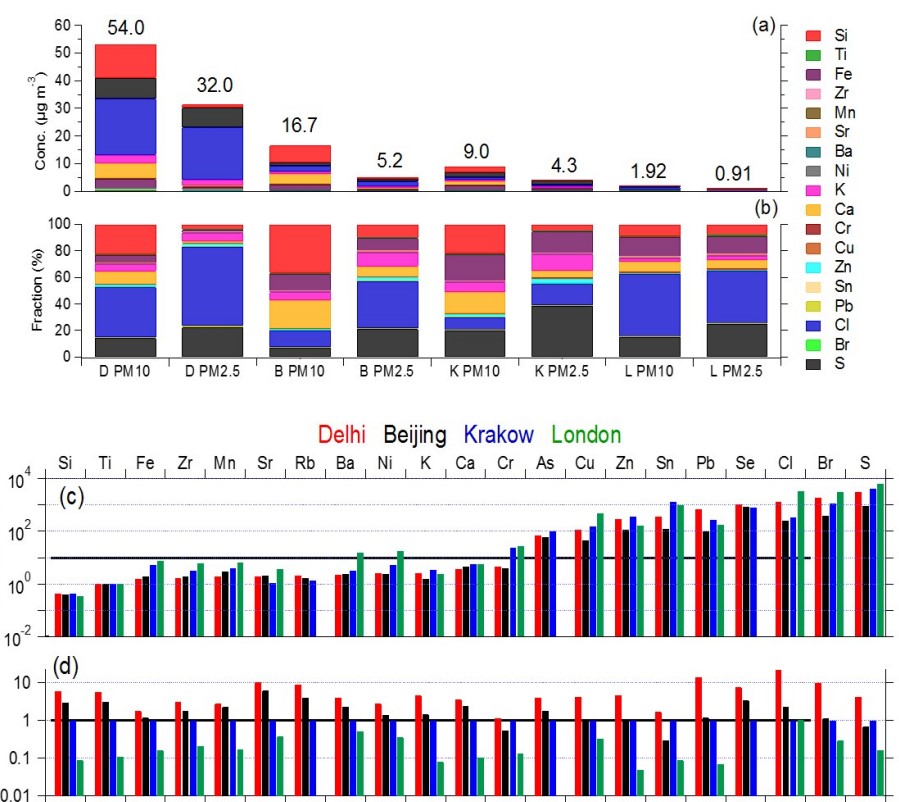

Figure 1: (a) Averaged elemental concentrations and (b) fractions (%) of elements in both size ranges at all four sites (Delhi (D),
Beijing (B), Krakow (K), London (L)); (c) Enrichment factors (using Ti as reference) of the measured elements in PM$_{10}$ (EF ~10
(solid line)); (d) averaged elemental concentrations in PM$_{10}$ normalized by those at Krakow. Note that Rb, As and Se are not included
in (a) and (b) because of absence in the London dataset, while all three are considered in (c) and (d) for the comparison between the
rest of the sites.

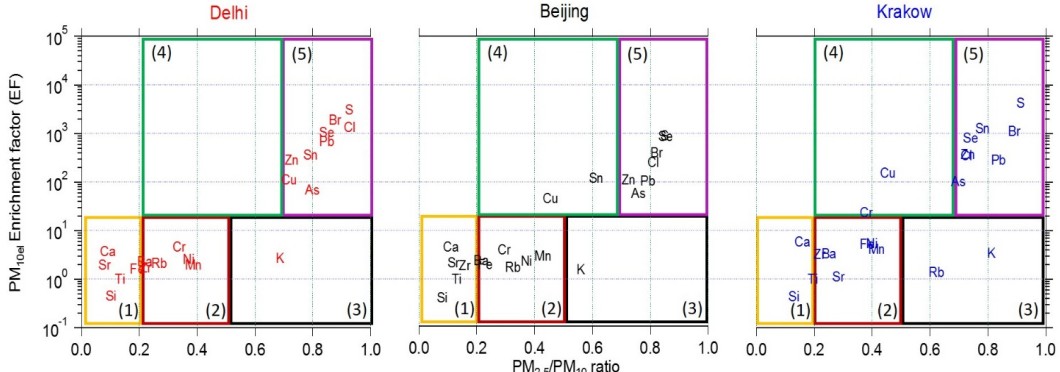

Figure 2: Classification of the measured elements in five groups for Delhi, Beijing, and Krakow based on their enhancement factor
(EF) vs PM$_{2.5}$/PM$_{10}$ values. PM$_{10}$ EF vs PM$_{2.5}$/PM$_{10}$ values and PM$_{2.5}$ EF vs PM$_{2.5}$/PM$_{10}$ values for all four sites are shown in
Supplementary (Fig. S5).





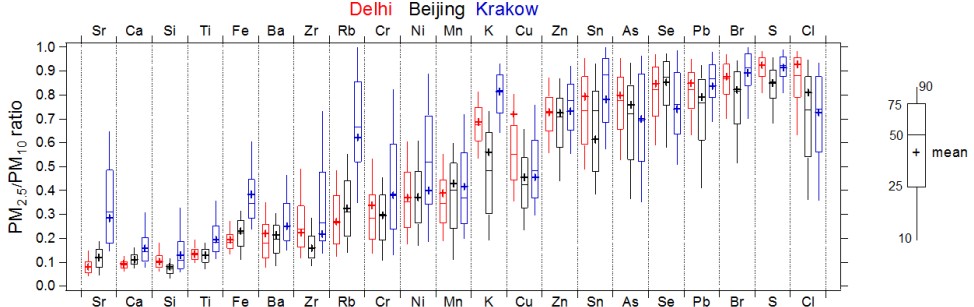

**Figure 3: Box-whisker plot of the measured elemental PM$_{2.5}$/PM$_{10}$ ratios at Delhi, Beijing, and Krakow (Fig. S4 is shown for all four**
**sites). Box: First to third quartile range, −: median line, +: mean, whiskers: 10-90% percentiles.**

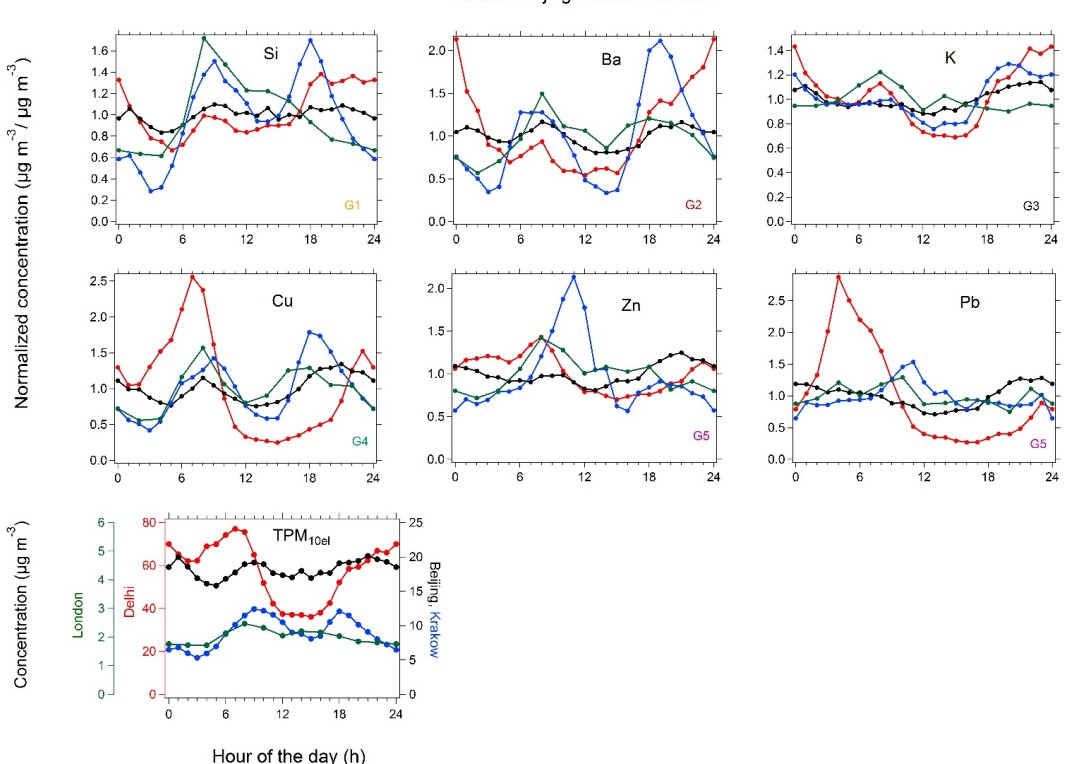

**Figure 4: Diurnal patterns (means) of selected elements representative of each group (G1: Group 1, G2: group 2, G3: Group 3, G4: Group 4, G5: Group 5) in PM$_{10}$ normalized by the mean values of the elements in PM$_{10}$, and the total elemental PM$_{10}$ (in µg m$^{-3}$, bottom) at all sites. Note that due to the time resolution of the original data the London data are 2-h averages, while the other data**
**are 1-h averages.**

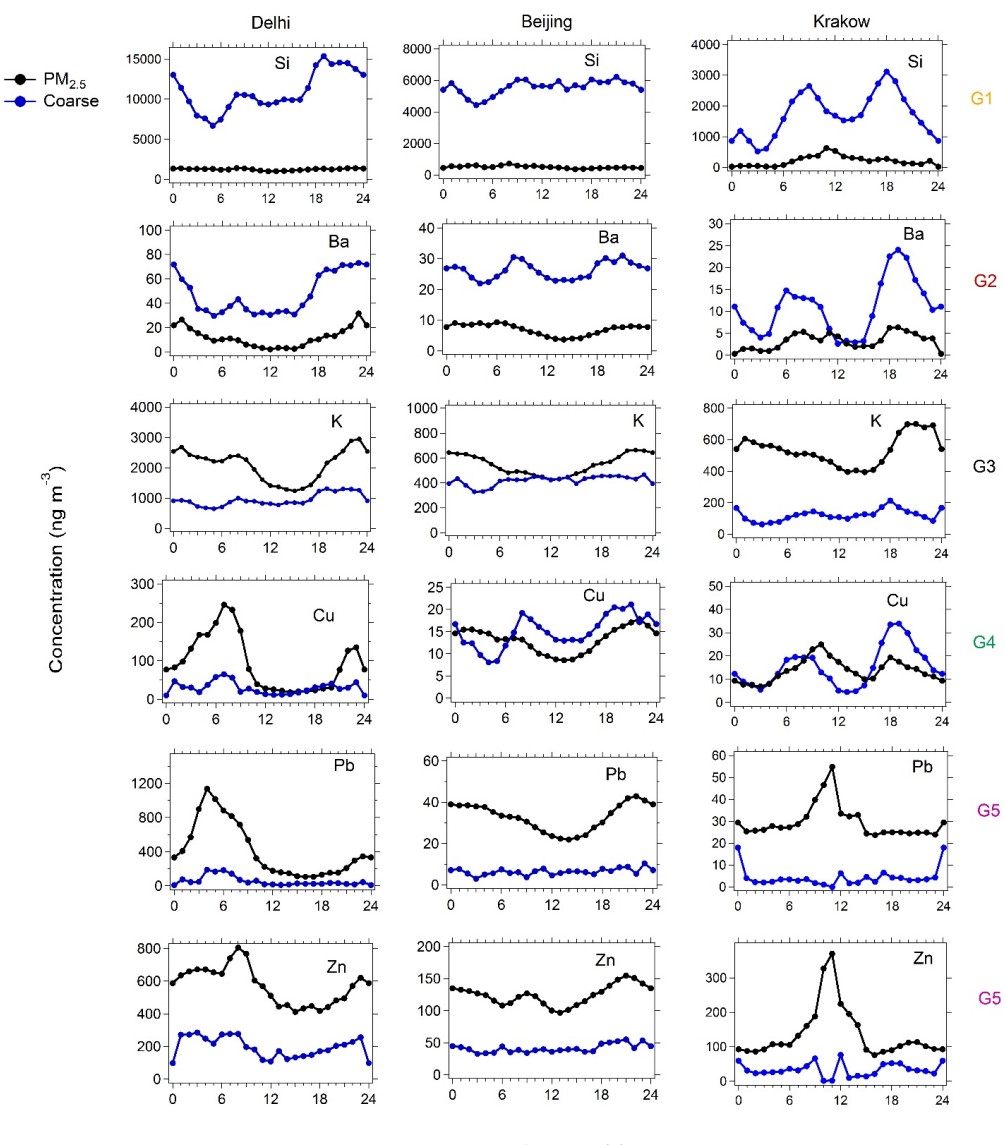

Figure 5: Diurnal variations of elements representative of each group (G1: Group 1, G2: group 2, G3: Group 3, G4: Group 4, G5: Group 5) in PM₂.₅ and coarse size fractions (PM₁₀–PM₂.₅) at Delhi, Beijing, and Krakow (see Fig. S8 for London).