# Peer review of "Highly time-resolved measurements of element concentrations in PM10 and PM2.5: Comparison of Delhi, Beijing, London, and Krakow"

_Atmospheric Chemistry and Physics, 2020_

## Referee Comment (RC1) · Anonymous Referee #1 · 9 Aug 2020

In this paper, size segregated (fine and coarse fraction) and highly time resolved (from 30 to 120 minutes) measurements of aerosol elemental composition are presented for some European and Asian cities (Krakow, London, Delhi, and Beijing). As outlined by the authors, the study of aerosol elemental composition is of interest since some elements contribute to adverse health effects (e.g. transition metals); moreover, elements are recognized as effective markers for source apportionment studies. Nevertheless, highly time resolved data - that are particularly important to trace source emissions - of the elemental composition given by online instrumentation (such as the Xact used in this work) are not widespread in the literature. Thanks to the high time resolution, average diurnal patterns of elements are presented and they can add interesting infor-

mation to the literature. The work is centered on the use of average crustal enrichment factors (EFs) and average PM2.5/PM10 ratios to derive qualitative information about emission processes. The method is not totally original, since it is well known that EFs and information about particle dimensions can be used to gain qualitative knowledge of natural/anthropogenic emission sources; anyway, its application to very different sites allows an interesting analysis of similarities and differences among them.

Since the method proposed aims to provide "a robust and useful framework for categorizing elements and assessing site-to-site differences" (line 282), my main concern regards the concept of "systematic shifts" (lines 273-278) for the Group 5 (and perhaps for Group 4, as reported in Section 4), that is not presented in a quantitative way, and I do not think it is very clear from Fig. 2. If the authors want to introduce this concept in the paper, it should be made more quantitative in order to be replicated in other papers in the future. For example, it would be better to underline the fact that the y-axes have a logarithmic scale, which means that graphical "vertical" shifts represent differences higher than the same graphical "horizontal" shifts. Then, differences in the values of PM10el EF for the same element for the different sites can be reported, in order to test if Group 5 present systematic higher differences.

Specific minor points to be addressed:

- Lines 29-30: I would suggest to move the sentence "Hourly maximum concentrations of [. . .] at the other sites." at the beginning of line 26, before introducing the methodology. Information on concentrations of Pb and Zn is not related to the methodology, and it sounds a bit confusing to me in this position.

- Line 38: the word "concentration" is missing after the parenthesis containing the definition of PM2.5.

- Lines 59-61: It is true that the methodology proposed in this paper does not require a full source apportionment (SA) analysis, but I think it is worth to specify that the full SA analysis is still necessary if more quantitative information (e.g. impact of each

source/category of sources) is desirable. For example, I suggest to modify the first part of the sentence: "When the aim of the analysis is not to obtain quantitative information, this method was proved particularly useful since it does not require a full source apportionment (SA) analysis [. . .]".

- Line 63: I think it is more appropriate to replace "campaigns" with "sampling sites".

- Lines 121-122: the comment about the good data quality sounds a bit redundant after all the explanation given about differences between the two instrumentations (Xact and AMS).

- Line 140: Since no statistics about the average values is reported in the text, I suggest to clarify that it can be found in Fig. S2a.

- Lines 142-143: Since Fig. 4 and S2b represent all the sites (not only Delhi), I suggest to underline it for sake of clarity at the beginning of the sentence, e.g. "For the four sites PM10 diurnal cycles, and PM10el and PM2.5el time series, are shown in Fig. 4 and S2b, respectively. The total PM10el (and PM2.5el) concentrations in Delhi [. . .]".

- Line 161: the part of the sentence "where EFs»1 indicate strong anthropogenic influence" should be moved before the introduction of the corresponding PM2.5 and PM10 ratios. Moreover, I suggest to delete "In addition" and to explain a bit better what type of information the PM2.5/PM10 ratio can give, e.g. "The mass ratio PM2.5/PM10 for an element gives rough indication of the particle size distribution, that reflects the corresponding emission processes and can provide insight into specific sources."

- Line 163: I would say "abrasion processes (e.g. mineral dust resuspension and brake wear).

- Lines 178-179: It is difficult to me to understand the use of "consistently" and "consistent" in this sentence. I think that it is correct to say that "Elements associated with this group are typically of crustal origin.".

- Line 181: "representative" is missing in the sentence: "Si is selected as the Group 1

representative element.".

- Lines 184-185: I agree that most of the dust deposited on the road surface derives from road abrasion and vehicle abrasion; but it is not clear to me why dust from construction activities or agricultural soil are also dominant. Is there any literature work highlighting this connection? Why should deposition from other activities (e.g. biomass burning) be less important?

- Lines 238-245: I think that this discussion should be supported by some literature works regarding the separation considered for elements present in primary components and elements present in secondary ones.

- Lines 246-249: The correlation between Xact Cl and AMS-derived Cl- is good (from Fig. S1, R = 0.97 and 0.98 for Beijing and Delhi, respectively), but for Beijing the absolute concentration values from Xact are clearly higher (slope of 1.9); this difference has been discussed in Section 2.2. I think that in this case these measurements are not enough to assure the lack of Cl from sea/road salt from. Please explain better the sentence "Because the kinetics of secondary aerosol [. . .] with the partial exception of Cl"; why are surface area and volume introduced, before speaking of PM2.5/PM10 ratio?

---

## Referee Comment (RC2) · Anonymous Referee #3 · 7 Oct 2020

The authors present highly time-resolved measurements of size-fractionated elements in four cities in Asia and Europe. The high time resolution and size-segregated elemental dataset are indeed a powerful tool to assess aerosol composition, sources, health effects in complex urban environments. However, this kind of studies are not widespread in the literature. The full source apportionment was already presented in other articles, but the authors present an interesting and simple approach for the analysis of the dataset which allows a first characterization of the major sources, site-to-site similarities or differences and the identification of key information required for efficient policy development. Therefore I suggest it for publication after minor revisions: L. 38: add concentration before above L. 77: The sampling period in Krakow is a little different

respect to the one in the other three sites; it should be taken into account. L. 130-132: Such a low EF for Si in all the sites is quite strange. Differences in the soil composition relative to the assumed values for the continental crust in all the sites does not seem to be a reasonable explanation. XRF is known not to be the best analytical technique to detect low Z-elements like Al or Si; probably Si is under-estimated by the instrument. The authors should add some comments. L. 141: This is not true for Krakow, see comment above. L. 181: "Si is selected as the Group 1 element", pleas add typical or representative element Conclusions: I think the information reported here are interesting, but the authors should stress the importance of a complete source apportionment to obtain a quantitative apportionment of the different sources.

---

## Author Comment (AC1) · 28 Oct 2020

**Response to the comments of anonymous referee #1**

We thank the referee for handling our paper carefully and for providing valuable comments. The corrections were implemented in the main text and can be distinguished with the "track changes" tool of MS-Word. We addressed all the comments (in italic typeset) and prepared a point-to-point response (in regular typeset). Changes to the manuscript are indicated in blue font. Please note that the line numbers are with reference to the submitted manuscript and not the revised manuscript.

*In this paper, size segregated (fine and coarse fraction) and highly time resolved (from 30 to 120 minutes) measurements of aerosol elemental composition are presented for some European and Asian cities (Krakow, London, Delhi, and Beijing). As outlined by the authors, the study of aerosol elemental composition is of interest since some elements contribute to adverse health effects (e.g. transition metals); moreover, elements are recognized as effective markers for source apportionment studies. Nevertheless, highly time resolved data - that are particularly important to trace source emissions -of the elemental composition given by online instrumentation (such as the Xact used in this work) are not widespread in the literature. Thanks to the high time resolution, average diurnal patterns of elements are presented and they can add interesting information to the literature. The work is centered on the use of average crustal enrichment factors (EFs) and average $PM_{2.5}/PM_{10}$ ratios to derive qualitative information about emission processes. The method is not totally original, since it is well known that EFs and information about particle dimensions can be used to gain qualitative knowledge of natural/anthropogenic emission sources; anyway, its application to very different sites allows an interesting analysis of similarities and differences among them.*

We kindly thank the referee for careful review and constructive comments, which we addressed as explained in the responses given below.

**Comment #1**
*Since the method proposed aims to provide "a robust and useful framework for categorizing elements and assessing site-to-site differences" (line 282), my main concern regards the concept of "systematic shifts" (lines 273-278) for the Group 5 (and perhaps for Group 4, as reported in Section 4), that is not presented in a quantitative way, and I do not think it is very clear from Fig. 2. If the authors want to introduce this concept in the paper, it should be made more quantitative in order to be replicated in other papers in the future. For example, it would be better to underline the fact that the y-axes have a logarithmic scale, which means that graphical "vertical" shifts represent differences higher than the same graphical "horizontal" shifts. Then, differences in the values of $PM_{10el}$ EF for the same element for the different sites can be reported, in order to test if Group 5 present systematic higher differences.*

We greatly appreciate the issue raised by Referee #1 concerning logarithmic scale for y-axes. We have modified the text from line 273 as below:

Site-to-site differences are also evident in the location of the elements within the Group 5 box in Fig. 2 (and Fig. S5). Systematic shifts are evident between Beijing (elements clustered to the lower left), Delhi (elements clustered to the upper right; note that two of the elements at the lower left are Cu and Zn, which require a significant shift towards the upper right to even be included in Group 5), and Krakow (intermediate). It is important to notice that the y-axes in Figs. 2 and S5 have a logarithmic scale, while x-axes have a linear scale, which indicates that the graphical vertical shifts represent higher differences than the same graphical horizontal shifts. The mean ($\pm$ standard deviation) $PM_{10el}$ EFs for Group 5 elements in Delhi, Beijing and Krakow are 1190 ($\pm$ 1017), 384 ($\pm$ 357) and 1021 ($\pm$ 1425), respectively.

***Specific minor points to be addressed:***

**Comment #2**

*- Lines 29-30: I would suggest to move the sentence "Hourly maximum concentrations of [. . .] at the other sites." at the beginning of line 26, before introducing the methodology. Information on concentrations of Pb and Zn is not related to the methodology, and it sounds a bit confusing to me in this position.*

Done.

**Comment #3**

*- Line 38: the word "concentration" is missing after the parenthesis containing the definition of $PM_{2.5}$.*

Done.

**Comment #4**

*- Lines 59-61: It is true that the methodology proposed in this paper does not require a full source apportionment (SA) analysis, but I think it is worth to specify that the full SA analysis is still necessary if more quantitative information (e.g. impact of each source/category of sources) is desirable. For example, I suggest to modify the first part of the sentence: "When the aim of the analysis is not to obtain quantitative information, this method was proved particularly useful since it does not require a full source apportionment (SA) analysis [. . .]".*

We have modified line 59 suggested by Referee #1. Thank you for the suggestion.

**Comment #5**

*- Line 63: I think it is more appropriate to replace "campaigns" with "sampling sites".*

Done.

**Comment #6**

*- Lines 121-122: the comment about the good data quality sounds a bit redundant after all the explanation given about differences between the two instrumentations (Xact and AMS).*

We have removed lines 121-122 from the main text.

**Comment #7**

*- Line 140: Since no statistics about the average values is reported in the text, I suggest to clarify that it can be found in Fig. S2a.*

We have modified line 140 as follows:

Total measured concentrations at Delhi (54 µg m$^{-3}$ in PM$_{10}$; 32 µg m$^{-3}$ in PM$_{2.5}$) are three times higher than those at the other sites, followed by Beijing (16.7 µg m$^{-3}$; 5.2 µg m$^{-3}$), Krakow (9 µg m$^{-3}$; 4.3 µg m$^{-3}$) and London (1.9 µg m$^{-3}$; 0.9 µg m$^{-3}$) (see Fig. S2a for average value statistics).

**Comment #8**

*- Lines 142-143: Since Fig. 4 and S2b represent all the sites (not only Delhi), I suggest to underline it for sake of clarity at the beginning of the sentence, e.g. "For the four sites PM10 diurnal cycles, and PM10el and PM2.5el time series, are shown in Fig. 4 and S2b, respectively. The total PM10el (and PM2.5el) concentrations in Delhi [. . .]".*

Done.

**Comment #9**

*- Line 161: the part of the sentence "where EFs»1 indicate strong anthropogenic influence" should be moved before the introduction of the corresponding PM2.5 and PM10 ratios. Moreover, I suggest to delete "In addition" and to explain a bit better what type of information the PM2.5/PM10 ratio can give, e.g. "The mass ratio PM2.5/PM10 for an element gives rough indication of the particle size distribution, that reflects the corresponding emission processes and can provide insight into specific sources."*

We have modified line 161 suggested by Referee #1.

**Comment #10**

*- Line 163: I would say "abrasion processes (e.g. mineral dust resuspension and brake wear).*

We modified it as "abrasion processes (e.g. mineral dust resuspension and brake/tire wear)".

**Comment #11**

*- Lines 178-179: It is difficult to me to understand the use of "consistently" and "consistent" in this sentence. I think that it is correct to say that "Elements associated with this group are typically of crustal origin."*

We have modified lines 178-179 suggested by Referee #1.

**Comment #12**

*- Line 181: "representative" is missing in the sentence: "Si is selected as the Group 1 representative element."*

Done.

**Comment #13**

*- Lines 184-185: I agree that most of the dust deposited on the road surface derives from road abrasion and vehicle abrasion; but it is not clear to me why dust from construction activities or agricultural soil are also dominant. Is there any literature work highlighting this connection? Why should deposition from other activities (e.g. biomass burning) be less important?*

Indeed, any anthropogenic and natural sources may result in the deposition of PM on the road surface. However, in this paragraph we reported elements with the lowest EFs (close to crustal origin) and the highest fraction of coarse particles, which are sufficiently large to settle out under gravity and deposit on the road surface. Whereas fine particles from high temperature emissions (e.g. biomass burning) can be less important in the deposition process. The composition of road dust has been found to be dominated by elements and compounds typically associated with road/vehicle abrasion and agricultural/construction soil dust (Thorpe and Harrison, 2008 and references therein).

We have cited the following reference in the main text at line 185:

Thorpe, A. and Harrison, R. M.: Sources and properties of non-exhaust particulate matter from road traffic: A review, Sci. Total Environ., 400, 270–282, https://doi.org/10.1016/j.scitotenv.2008.06.007, 2008.

**Comment #14**

*- Lines 238-245: I think that this discussion should be supported by some literature works regarding the separation considered for elements present in primary components and elements present in secondary ones.*

We have cited the following references:

Seinfeld, J. H. and Pandis, S. N.: Atmospheric chemistry and physics: from air pollution to climate change, Eds. 2, John Wiley & Sons, Inc., New York, USA, 2006.

Liu, L., Kong, S., Zhang, Y., Wang, Y., Xu, L., Yan, Q., Lingaswamy, A. P., Shi, Z., Lv, S., Niu, H., Shao, L., Hu, M., Zhang, D., Chen, J., Zhang, X., and Li, W.: Morphology, composition, and mixing state of primary particles from combustion sources—crop residue, wood, and solid waste, Sci. Rep., 7, 1–15, https://doi.org/10.1038/s41598-017-05357-2, 2017.

Zhang, R., Jing, J., Tao, J., Hsu, S.-C., Wang, G., Cao, J., Lee, C. S. L., Zhu, L., Chen, Z., Zhao, Y., and Shen, Z.: Chemical characterization and source apportionment of $PM_{2.5}$ in Beijing: seasonal perspective, Atmos. Chem. Phys., 13, 7053–7074, https://doi.org/10.5194/acp-13-7053-2013, 2013.

**Comment #15**

*- Lines 246-249: The correlation between Xact Cl and AMS-derived Cl- is good (from Fig. S1, R = 0.97 and 0.98 for Beijing and Delhi, respectively), but for Beijing the absolute concentration values from Xact are clearly higher (slope of 1.9); this difference has been discussed in Section 2.2. I think that in this case these measurements are not enough to assure*

*the lack of Cl from sea/road salt. Please explain better the sentence "Because the kinetics of secondary aerosol [. . .] with the partial exception of Cl"; why are surface area and volume introduced, before speaking of PM2.5/PM10 ratio?*

We have discussed the higher slope issue for Cl in Beijing in Section 2.2. We repeat here with additional explanation for higher slope and the lack of Cl from sea/road salt.

First, relative ionization efficiency for AMS measurements of Cl was not determined in Beijing (whereas calibrations with $NH_4Cl$ were performed in Delhi). Second, the interquartile range of Cl $PM_{2.5}/PM_{10}$ at Beijing is quite wide (0.5 to 0.9), with the lower values approximately matching those of Zn and Pb and suggesting that primary emissions of $ZnCl_2$ and $PbCl_2$ are important at this site, which are not efficiently detected in standard AMS operation.

High Cl concentrations from November to March in Beijing are reported in previous studies, which is believed to be associated with coal burning (Yao et al., 2002; Zhang et al., 2019). The contribution from sea-salt particles is less important because the sampling site in Beijing is about 200 km from the sea. In addition to that, backward trajectories were calculated for the sampling period (Rai et al., in prep), which also brackets the absence of air masses from the coastal oceans. However, the sea/road-salt discussion would be strengthened by the measurement of Na, which is an important tracer of sea/road-salt in the form of NaCl. While Na and Cl are good tracers for sea/road-salt, the Cl/Na ratio in Beijing during winter is reported to be higher (2.3) than the ratio in seawater (1.17) (Yao et al., 2002).

We have modified Section 2.2 as follows:

The Xact/AMS ratio for Cl observed in Beijing likely occurs because the relative ionization efficiency for AMS measurements of Cl was not determined in Beijing (whereas calibrations with $NH_4Cl$ were performed in Delhi). In addition, the Beijing measurements likely have a higher fraction of other forms of Cl (e.g. $ZnCl_2$, $PbCl_2$, $FeCl_3$), which are not efficiently detected in standard AMS operation. High Cl concentrations from November to March in Beijing are reported in previous studies, which is believed to be associated with coal burning (Yao et al., 2002; Zhang et al., 2019). The contribution from sea-salt particles is less important because the sampling site in Beijing is about 200 km from the sea. However, the sea/road-salt discussion would be strengthened by the measurement of Na, which is an important tracer of sea/road-salt in the form of NaCl. While Na and Cl are good tracers for sea/road-salt, the Cl/Na ratio in Beijing during winter is reported to be much higher (2.3) than the ratio in seawater (1.17) (Yao et al., 2002).

We have rephrased the line 248 as follows:

The $PM_{2.5}/PM_{10}$ of these elements is among the highest recorded, with the partial exception of Cl, which is probably due to the fact that secondary aerosol condensation is driven by surface area rather than volume.

Yao, X. H., Chan, C. K., Fang, M., Cadle, S., Chan, T., Mulawa, P., He, K. B., and Ye, B. M.: The water-soluble ionic composition of $PM_{2.5}$ in Shanghai and Beijing, China, Atmos. Environ., 36, 4223–4234, https://doi.org/10.1016/S1352-2310(02)00342-4, 2002.

Zhang, B., Zhou, T., Liu, Y., Yan, C., Li, X., Yu, J., Wang, S., Liu, B., and Zheng, M.: Comparison of water-soluble inorganic ions and trace metals in $PM_{2.5}$ between online and offline measurements in Beijing during winter, Atmos. Pollut. Res., 10, 1755-1765, 10.1016/j.apr.2019.07.007, 2019.

---

## Author Comment (AC2) · 28 Oct 2020

**Response to the comments of anonymous referee #2**

We thank the referee for handling our paper carefully and for providing valuable comments. The corrections were implemented in the main text and can be distinguished with the "track changes" tool of MS-Word. We addressed all the comments (in italic typeset) and prepared a point-to-point response (in regular typeset). Changes to the manuscript are indicated in blue font. Please note that the line numbers are with reference to the submitted manuscript and not the revised manuscript.

*The authors present highly time-resolved measurements of size-fractionated elements in four cities in Asia and Europe. The high time resolution and size-segregated elemental dataset are indeed a powerful tool to assess aerosol composition, sources, health effects in complex urban environments. However, this kind of studies are not widespread in the literature. The full source apportionment was already presented in other articles, but the authors present an interesting and simple approach for the analysis of the dataset which allows a first characterization of the major sources, site-to-site similarities or differences and the identification of key information required for efficient policy development. Therefore I suggest it for publication after minor revisions:*

We kindly thank the referee for taking our manuscript into consideration and we value the comments raised to improve the manuscript. A point-to-point response to the issues raised is enclosed below.

**Comment #1**
*L. 38: add concentration before above.*

Done.

**Comment #2**
*L. 77: The sampling period in Krakow is a little different respect to the one in the other three sites; it should be taken into account.*

We have added the following text at L. 76:

It is important to notice that the sampling period in Krakow is different from the rest of the sites.

**Comment #3**
*L. 130-132: Such a low EF for Si in all the sites is quite strange. Differences in the soil composition relative to the assumed values for the continental crust in all the sites does not seem to be a reasonable explanation. XRF is known not to be the best analytical technique to detect low Z-elements like Al or Si; probably Si is under-estimated by the instrument. The authors should add some comments.*

We appreciate the concern raised by Referee#2. We have modified L. 130-132 as follows:

The unexpectedly low EFs observed for Si (0.41–0.45) and compared to previous studies (Majewski and Rogula-Kozłowska, 2016; Tao et al., 2013), are likely due to self-attenuation issues in XRF analysis for lighter elements (atomic number<19), which may cause underestimation in their concentrations (Maenhaut et al., 2011; Visser et al., 2015). Therefore, the measurements of Al and Si from Xact need to be treated with caution. However, low EFs for Si is also probably due to crust-air fractionation in the wind-blown generation of crustal aerosol particles (Rahn, 1976).

Maenhaut, W., Raes, N., and Wang, W.: Analysis of atmospheric aerosols by particle induced X-ray emission, instrumental neutron activation analysis, and ion chromatography, Nucl. Inst. Meth. Phys. Res. Sect. B: Beam Interact. Mater. Atoms., 269, 2693–2698, https://doi.org/10.1016/j.nimb.2011.08.012, 2011.

Visser, S., Slowik, J. G., Furger, M., Zotter, P., Bukowiecki, N., Dressler, R., Flechsig, U., Appel, K., Green, D. C., Tremper, A. H., Young, D. E., Williams, P. I., Allan, J. D., Herndon, S. C., Williams, L. R., Mohr, C., Xu, L., Ng, N. L., Detournay, A., Barlow, J. F., Halios, C. H., Fleming, Z. L., Baltensperger, U., and Prévôt, A. S. H.: Kerb and urban increment of highly time-resolved trace elements in $PM_{10}$, $PM_{2.5}$ and $PM_{1.0}$ winter aerosol in London during ClearfLo 2012, Atmos. Chem. Phys., 15, 2367–2386, https://doi.org/10.5194/acp-15-2367-2015, 2015.

Rahn, K. A.: Silicon and aluminum in atmospheric aerosols: crust air fractionation?, Atmos. Environ., 10, 597–601, https://doi.org/10.1016/0004-6981(76)90044-5, 1976.

**Comment #4**
*L. 141: This is not true for Krakow, see comment above.*

We have modified L. 141 as follows:

Although the measurement periods do not overlap, they were all performed during the colder months of the year (partially true for Krakow, see Section 2.1), and characteristic features of each site are evident.

**Comment #5**
*L. 181: "Si is selected as the Group 1 element", pleas add typical or representative element.*

Done.

**Comment #6**
*Conclusions: I think the information reported here are interesting, but the authors should stress the importance of a complete source apportionment to obtain a quantitative apportionment of the different sources.*

We have added the following text on L. 315:

Although the method proposed in this work allows for a comparison of the characteristics in different cities, a full SA analysis is necessary if more quantitative information (e.g. source contributions) is desired.

---

## Author Response (AR2)

**Comments to the Author:**

*The authors have appropriately addressed the comments of the two anonymous referees and they have modified their manuscript accordingly. However, a number of alterations and corrections are needed for the Main text and Supplement before the manuscript can be published in ACP.*

Dear Prof. Dr. Maenhaut,

Thank you very much for giving us this opportunity to resubmit our final version of the manuscript. We addressed all comments (in italic typeset) and prepared a point-to-point response (in regular typeset). In the following, page and lines references refer to the previous version of the manuscript reviewed by Editor.

*For the Main text:*

*Line 24: Replace "in Asian" by "in the Asian".*

Done.

*Line 61: The acronym "SA" is already define in line 45; therefore, replace "source apportionment (SA) analysis" by "SA analysis".*

Done.

*Line 75: Replace "the zinc" by "a zinc".*

Done.

*Line 78: Replace "to notice" by "to note".*

Done.

*Line 109: Replace "to other" by "to the other".*

Done.

*Lines 124, 330, and 333: Replace "e.g." by "e.g.,".*

Done.

*Line 171: The acronym "EF" is already define in line 131; therefore, replace "enrichment factor (EF)" by "EF".*

Done.

*Line 175: Replace "results in" by "result in".*

Done.

*Line 186: Replace "concentrations" by "concentrations of".*

Done.

*Line 190: Replace "and Fe" by "Fe".*

Done.

*Line 246: Replace "median and" by "medians and".*

Done.

*Line 255: Replace "components elements" by "component elements".*

Done.

*Line 285: Replace "approximately" by "at approximately".*

Done.

*Line 290: Replace "to notice" by "to note" and replace "while x-axes" by "while the x-axes".*

Done.

*Line 338: Replace "AS, JN" by "AS, and JN".*

Done.

*Line 405: Replace "5," by "5, 100065,".*

Done.

*Line 413: Replace "2012, 13" by "2012, 585791".*

Done.

*Line 484: Replace "2013, 15" by "2013, 942916".*

Done.

*Line 508: Replace "742," by "742, 140332,".*

Done.

*Line 528: Replace "228: 290" by "228, 290".*

Done.

*Line 616: Replace "their enrichment" by "their PM10 enrichment".*

Done.

*Line 620: Replace "plot of" by "plots of" and replace "Fig. S4 is shown" by "see Fig. S4".*

Done.

*For the Supplement:*

*Page 3, line 8: Replace "plot of" by "plots of".*

Done.

*Pages 12-13, Table S2: Many numeric data in the Table are given with too many significant figures; two significant figures normally suffice and three suffice when the first significant figure is "1"; thus, as examples, replace "3089" by 3100" and replace "12158" by "12200".*

Done. We have also modified Tables S1 and S3 accordingly.

[revised manuscript text omitted]